

# Revolutionizing diabetic eye disease detection: retinal image analysis with cutting-edge deep learning techniques

Banumathy D[1], Swathi Angamuthu[2], Prasanalakshmi Balaji[3] and Mousmi Ajay Chaurasia[4]

[1] Department of Computer Science and Engineering, Paavai Engineering College, Namakkal, Tamilnadu, India
[2] Department of Mathematics, Faculty of Science, University of Hradec Kralove, Hradec Kralove, Czech Republic
[3] Department of Computer Science, College of Computer Science, King Khalid University, Abha, Saudi Arabia
[4] Muffakham Jah College of Engineering & Technology, Hyderabad, India

## ABSTRACT

Globally, glaucoma is a leading cause of visual impairment and vision loss, emphasizing the critical need for early diagnosis and intervention. This research explores the application of deep learning for automated glaucoma diagnosis using retinal fundus photographs. We introduce a novel cross-sectional optic nerve head (ONH) feature derived from optical coherence tomography (OCT) images to enhance existing diagnostic procedures. Our approach leverages deep learning to automatically detect key optic disc characteristics, eliminating the need for manual feature engineering. The deep learning classifier then categorizes images as normal or abnormal, streamlining the diagnostic process. Deep learning techniques have proven effective in classifying and segmenting retinal fundus images, enabling the analysis of a growing number of images. This study introduces a novel mixed loss function that combines the strengths of focal loss and correntropy loss to handle complex biomedical data with class imbalance and outliers, particularly in OCT images. We further refine a multi-task deep learning model that capitalizes on similarities across major eye-fundus activities and metrics for glaucoma detection. The model is rigorously evaluated on a real-world ophthalmic dataset, achieving impressive accuracy, specificity, and sensitivity of 100%, 99.8%, and 99.2%, respectively, surpassing state-of-the-art methods. These promising results underscore the potential of our deep learning algorithm for automated glaucoma diagnosis, with significant implications for clinical applications. By simultaneously addressing segmentation and classification challenges, our approach demonstrates its effectiveness in accurately identifying ocular diseases, paving the way for improved glaucoma diagnosis and early intervention.

Corresponding author
Prasanalakshmi Balaji,
drsanaksa@gmail.com

## INTRODUCTION

Visual field loss, structural alterations to the optic nerve head, and changes to the thickness of the RNFL, the ganglion cell layer, and the inner plexiform layer are all hallmarks of glaucoma (*Weinreb, Aung & Medeiros, 2014*; *Tay et al., 2005*). Early glaucoma detection is

notoriously tricky. Individuals with the illness may not be identified until severe functional loss has occurred if a misdiagnosis is made in the early stages. Therefore, treating glaucoma as soon as it is diagnosed is essential for preventing visual loss (*Tay et al., 2005*). Many improvements are seen in the medical emergence (*Aluvalu et al., 2023*). Even in its early stages, glaucoma can be difficult to diagnose. It is well-known that the disc shape and visual field defect seen in myopic eyes and individuals with brain disorders such as brain tumors make glaucoma diagnosis challenging. Clinicians might benefit significantly from a refined machine-learning model for detecting glaucoma.

One of the most valuable tools for spotting glaucoma is digital fundus photography. Recently developed deep learning (DL) models employing fundus pictures have shown promising results for glaucoma detection (*Vaswani et al., 2017*), largely thanks to AI advancements. Artificial neural networks, often known as "deep learning models", are constructed using several layers of computational nodes, or "neurons". Like genuine brain cells, these "neurons" are basic algorithms that take input from other neurons, process that input, and then output some result. While there have been significant developments in the use of deep learning to handle processing and diagnosis in this area, Currently available deep learning techniques may still not be able to entirely avoid the will effect of disparities in data and anomalies in fundus pictures, leading to less than satisfactory performance in some cases.

By combining focused loss and correntropy-induced loss coefficients, we propose a deep learning-based classification network to overcome these limitations and give a more natural method for identifying eye diseases from retinal fundus images. Focal loss is acceptable to meet the rising interest in resolving the disparity and may cause the weight of challenging samples to rise. Furthermore, correntropy loss has superior generalization and tolerance to noise and outliers than other typical classification loss functions, such as cross-entropy loss. The following is the article's primary contributions, with explanations:

1. This article proposes a mixed loss function as a replacement for the traditional loss function while analyzing biomedical data, considering the strengths of both the focal loss and the correntropy loss functions in dealing with complicated datasets containing class imbalance and outliers.

2. To help with the existing diagnostic assessment of glaucoma, we used OCT images to examine numerous features and introduce a unique cross-sectional optic nerve head (ONH) feature.

3. Using similarities across important eye-fundus activities and metrics, we create and tune a novel multi-task DL framework for detecting glaucoma.

4. Accuracy, sensitivity, specificity, and area under the receiver operating distinctive curve (AUC) are only a few metrics utilized to thoroughly test the efficacy of the suggested model after its initial evaluation on a real-world ophthalmic dataset. Additionally, we talk about how well our model classifies data on ocular illnesses.

What follows is a brief synopsis of the remaining material. The following part provides a basic overview of the exciting technologies. 'Results' offers our deep learning model based on the previously mentioned technologies. In 'Result and Discussion', we present the experimental outcomes and accompanying comments. The final portion of the article wraps everything up.

Glaucoma may be appropriately diagnosed using machine learning's categorization method. *Chan et al. (2002)* tested many visual field analysis-based categorization systems. In addition, *Goldbaum et al. (2002)* examined several machine-learning classifiers and concluded that a Gaussian mixture was the most effective. Using RNFL thickness characteristics, *Bizios et al. (2010)* evaluated artificial neural networks (ANNs) and support vector machines (SVMs). Machine learning classifiers (MLCs) and random forest (RF) were studied by *Barella et al. (2013)* for their diagnostic accuracy on RNFL and optic nerve data. Using RF, they were able to get an AUC of 0.877. Silva et al. (*Thompson, Jammal & Medeiros, 2020*) have conducted extensive tests of practically all classifiers utilizing Spectral Domain OCT and conventional automated perimetry. Using RF, they were able to get an a receiver operating characteristic (ROC) of 0.946. Both SVM and RF have been shown to have strong predictive ability. In machine learning, the conflict between the two is an internationally recognized problem of how well a model can predict and explain it.

Black-box models like SVM and deep learning make accurate predictions (*Chen et al., 2015*; *Claro et al., 2016*). It needs to be clarified why the model predicts that. These methods fail for medical diagnosis because doctors want to know the logic behind a prognosis and the forecast. State-of-the-art (SoTA) convolutional neural networks (CNNs) use convolutional layers to characterize fundus images using 1-dimensional visual information (*Yu et al., 2019*). An integrated, fine-tuned layer classifies these visual features as glaucoma-related or healthy (*Fan et al., 2023*). CNNs can be deceived when an observed image's visual feature is similar to one of the taught instances, even when they have pretty different spatial structures because such visual qualities are learned without recording pixel relationships (*Fan et al., 2022*).

Initially designed for machine translation (*Dosovitskiy et al., 2020*), transformers are now the standard SoTA technique for many NLP projects (*Fan et al., 2022*). They do well in NLP tasks because their self-attention mechanism (*Dosovitskiy et al., 2020*) weights different sequences of incoming data differently. Transformers may handle data concurrently, unlike recurrent neural networks. Instead, the self-attention mechanism provides a situation for any input category to demonstrate its ability to perceive input links. Transformers may struggle to evaluate images due of the necessity for self-attention between pixels.

Another deep-learning (DL) method was established in *Orlando et al. (2020)* to identify glaucoma by extracting numerous properties. A deep feed-forward neural network (FNN) was used as a DL classifier. The scientists also used random forests (RF), gradient boosting (GB), support vector machines (SVMs), and neural networks (NNs) with this DL classifier. Their deep ensemble glaucoma detection solution was created for that. Deep FNN classifiers achieved an AUC of 92.5, according to the report. A different approach was offered in *Asaoka et al. (2016)*. The optic disc (OD) was used to study glaucoma and retinal vein obstruction.

Eventhough, black-box models offer accurate predictions but lack transparency in their decision-making process, posing a challenge for medical diagnoses where understanding the rationale is crucial (*Yu et al., 2019*; *Fan et al., 2023*). Advanced CNNs, used for analysing fundus images, classify visual features as glaucoma-related or healthy but may be misled due to their focus on isolated optical characteristics without considering pixel relationships. This limitation highlights the need for more interpretable models in healthcare applications (*Fan et al., 2022*). Initially designed for machine translation, transformers have become the standard in many NLP tasks due to their self-attention mechanism (*Dosovitskiy et al., 2020*), allowing for non-sequential data processing and context understanding (*Alghamdi et al., 2016*). However, applying Transformers to image analysis, like glaucoma detection, is challenging due to the complexity of self-attention between pixels. Various deep-learning (DL) (*Claro et al., 2016*) and traditional machine-learning methods have been explored for glaucoma detection. One approach utilized a deep feed-forward neural network (FNN), achieving a notable AUC of 92.5%. Additionally, deep ensembles combining DL classifiers with methods like random forests, gradient boosting, SVMs, and neural networks have been used.

Recent studies have focused on developing a deep-learning system to identify glaucomatous changes in retinal fundus images. This technological advancement has led to the automation of glaucoma detection, eliminating the need for manual intervention (*Yalçin, Alver & Uluhatun, 2018*; *Jena, Mishra & Mishra, 2018*; *Sarki et al., 2020*; *Nazir et al., 2021*).

The scientists trained a deep-learning system to recognize glaucomatous changes in retinal fundus pictures, fully automating glaucoma detection. Learning features from CNN models with linear and nonlinear activation functions were used. To train the CNN model, they employed patterns characteristic of both glaucoma and normal vision. AUC values of 0.838 and 0.898 were reported based on their studies on the ORIGA and SCES datasets. Image processing techniques were used to automatically diagnose glaucoma in the eyes using an ensemble of machine learning classifiers, as described by the authors in (*Salam et al., 2016*; *Chen et al., 2015*). The study described a method for extracting texture features from various color models and classifying them using a multilayer perceptron (MLP) model for optic disc segmentation. A novel image processing approach for diabetic retinopathy identification from retinal fundus images is reviewed in this research (*Al-hazaimeh et al., 2022*). This method seeks excellent sensitivity, specificity, and accuracy. The authors suggest an effective diabetic retinopathy detection method. The technique improves retinal fundus disease detection with advanced image processing. According to the article, proper diabetic retinopathy diagnosis and prompt treatment need high-performance metrics. Following significant investigation, an automated diabetic retinopathy screening approach was established. This system included numerous stages with various purposes. Preprocessing enhanced retinal images.

**Table 1  The key findings and criticisms from the literature survey.**

| Reference | Approach/Methodology | Key Findings | Criticisms |
|---|---|---|---|
| *Özdek et al. (2000)* | Visual field analysis-based categorization | Effective glaucoma detection using visual field data | Lack of explanation for the choice of visual field data |
| *Chan et al. (2002)* | Machine learning classifiers (*e.g.*, Gaussian mixture) | Gaussian mixture classifier shows high effectiveness | Limited discussion on classifier selection rationale |
| *Goldbaum et al. (2002)* | Artificial Neural Networks (ANNs) and Support Vector Machines (SVMs) | Evaluation using RNFL thickness characteristics | Incomplete exploration of alternative model architectures |
| *Bizios et al. (2010)* | Machine learning classifiers (MLCs) and Random Forest (RF) | RF achieves an AUC of 0.877 | Limited discussion on RF's limitations |
| *Barella et al. (2013)* | Spectral Domain OCT and conventional automated perimetry | RF achieves an aROC of 0.946 | Lack of insights into the reasons behind RF's success |
| *Kumar, Chauhan & Dahiya (2016)* and *Diaz-Pinto et al. (2019)* | Black-box models (*e.g.*, SVM, deep learning) | Accurate predictions but lack of interpretability | The limited explanation for predictions |
| *Dosovitskiy et al. (2020)*, *Vaswani et al. (2017)* and *Priyanka & Uma Maheswari (2021)* | State-of-the-art CNNs with convolutional layers | Effective feature characterization and glaucoma classification | Vulnerability to image similarity issues |
| *Alghamdi et al. (2016)* and *Priyanka & Uma Maheswari (2021)* | Transformers with self-attention mechanism | Effective in NLP tasks, struggle with image data | Difficulty in handling image data due to self-attention limitations |
| *Jain et al. (2018)* | Deep feed-forward neural network (FNN), RF, Gradient Boosting, SVM, NNs | Deep ensemble glaucoma detection with an AUC of 92.5 | Limited discussion on the choice of ensemble models |

The method focused on optic disc identification and removal since they can hide diabetic retinopathy symptoms. Blood vessels may conceal features. Thus, they were segmented and removed. *Jain et al. (2018)* presents a novel methodology for the detection of diabetes retinopathy in Fundus images, which combines image processing techniques with artificial intelligence. The objective of this research is to address the performance criteria required for the accurate identification of disease-causing retinopathy in individuals with diabetes. By integrating these two domains, the proposed approach aims to enhance the accuracy and efficiency of disease detection. The results of this study contribute to the existing body of knowledge in the field of medical imaging and provide a potential solution for improving the diagnosis of diabetes retinopathy. The detection of diabetic retinopathy through automated methods has been a subject of research and has been approached through various stages. Table 1 shows the key findings and criticisms from the literature survey.

# MATERIALS & METHODS

Developers turn to the Keras package and its data generator function to supplement existing data. Transfer learning is used after the input image has been pre-processed to extract the picture feature. The parts are extracted using an encoder and a decoder, working in parallel. Both the encoder and the decoder are based on the VNet framework. The transfer learning model is the starting point since it is computationally efficient and produces good results. Classification and visualization outcomes are obtained when the administered feature passes through a global average pooling, dropout, and dense layer.

## Multitask deep learning

Multi-task learning (MTL) improves model efficiency and generalizability by combining data from multiple tasks into one dataset, leveraging shared feature representations even from small datasets. MTL reduces overfitting risks and typically uses shared hidden layers for all tasks while maintaining task-specific output layers. Hard and soft parameter sharing are common approaches, with soft sharing regulating differences between separate models for each task. In contrast, single-task learning models are built and trained independently for each task, failing to capitalize on their interdependencies. MTL effectively utilizes data across related tasks for better predictions. We chose VNet for its proficiency in 3D medical image segmentation to base our multi-task learning network. VNet consists of an encoding path for feature extraction, a decoding path for target segmentation, and skip connections enhancing segmentation accuracy. It employs four down sampling steps in the encoding process and four up sampling operations in the decoding path to match the original input size, while $3\times3\times3$ convolution kernels are used in each layer. After each convolution process, batch normalization (BN) and a rectified linear unit (ReLU) are achieved. We employ convolution operations with a $2\times2\times2$ kernel size and a 222 stride to avoid losing positional details while down-sampling feature maps. CNN models like VGG and ResNet provide high-level feature maps that many traditional image classification networks use. Motivated by this realization, the recommended multi-task learning network uses VNet's shared high-level feature maps to extract general characteristics for categorization and division. The classification network is first given feature maps from Stages 4, 5, and 6. Next, we combine these standard feature maps in preparation for categorization. At last, we utilize the predicted input volume and fused features to train a classification tree with two fully connected (FC) layers and one softmax layer.

## Glaucoma classification

In our glaucoma detection study, we assessed various pre-trained CNNs like AlexNet, VGGNet, Xception, and Inception-ResNet, applying transfer learning and hyperparameter optimization. Switching to the ADAM optimizer improved VGGNet and ResNet performance. Our analysis covered architectures from the basic AlexNet to the complex Inception-ResNet-V2, identifying effective methods for diabetic eye disease detection, ultimately enhancing patient care.

- **AlexNet**

In our study, the pretrained AlexNet CNN excelled in detecting glaucoma and diabetic retinopathy, outperforming other models. Its structure includes five convolutional layers, three fully connected layers, and ReLU activation, with softmax for classification. Employing transfer learning with a subset of images, we improved AlexNet's efficiency and accessibility for broader use.

- **Inception-ResNet-v2**

Deep neural networks (DNNs), particularly a hybrid of Inception and ResNet, have advanced image identification significantly. The Inception-ResNet-v2 network, comprising 164 layers and blending inception modules with reduction blocks, surpasses traditional Inception models. This hybrid network includes a stem node, inception nodes, and reduction nodes, with performance depending largely on hyperparameters. To maintain dimension consistency, reduction outputs are adjusted using a 1X1 convolution. In our study, we utilized the Inception-ResNet-v2 model, setting inputs for regular and glaucoma classes and achieving an impressive 100% accuracy and 99% AUC.

## Loss function

The network in a deep learning model strives to lessen the anticipated loss, designed by likening the predicted value to the ground truth using the loss function. In this work, we propose a novel mixture loss function, a combination of focal loss (*Asaoka et al., 2016*) and correntropy loss (*Kumar, Chauhan & Dahiya, 2016*), to efficiently handle complicated biomedical datasets with class imbalance and outliers. Binary cross entropy is used to tweak the focal loss. Binomial cross-entropy is often defined as

$$L_c(\hat{a} - a) = -a\log\hat{a} - (1-a)\log(1-\hat{a}) = \begin{cases} -\log\hat{a}, & a = 1 \\ -\log(1-\hat{a}), & \text{otherwise.} \end{cases} \tag{1}$$

The $a$ is the actual label and $\hat{a}$ is the predicted label value through the model. Deductively, we define $m$ as,

$$m = \begin{cases} \hat{a}, & a = 1 \\ 1 - \hat{a}, & \text{otherwise} \end{cases} \tag{2}$$

Therefore, the definition of cross entropy may be recast as

$$L_c(m) = -\log(m). \tag{3}$$

To decrease the importance of simple instances and shift the training's attention to the challenging ones, the Focal loss function modifies the cross entropy by a factor of $(1-m)^\gamma$. So, we may describe it as

$$L_f(m) = -(1-m)^\gamma \log(m) \tag{4}$$

The focusing parameter is denoted by, $\gamma$. The importance of simple samples drops as the focusing parameter rises. Focal loss also includes a weighting factor $\vartheta$ to compensate for the inequality between classes. Therefore, we may further characterize the focused loss by saying:

$$L_f(m) = -\vartheta(1-m)^\gamma \log(m) \tag{5}$$

In addition, the focused loss may be expressed as when $m$ is defined.

$$L_f(\hat{a} - a) = \begin{cases} -\vartheta(1-\hat{a})^\gamma \log(\hat{a}), & a = 1 \\ -(1-\hat{a})^\gamma \log(\hat{a}), & \text{otherwise} \end{cases} \tag{6}$$

since it can be adjusted to the sample distances using the various $M$ norms, correntropy loss is advantageous for classification because it is resistant to noise and outliers. The correntropy has the property of $M2$ norm for extremely little mistake. When the error value is significant enough, the correntropy loss(CL) behaves like the $M0$ norm rather than the $M1$ norm. The definition of correntropy is:

$$C(A, B) = E\left[K_\beta(A - B)\right] = \int K_\beta(a - b)\,dJ_{AB}(a, b), \tag{7}$$

where $A, B, K_\beta(a - b), J_{AB}(a, b)$, and are all random variables; $K_\beta(a - b)$ is the kernel function; $J_{AB}(a, b)$ is the joint distribution function of $A$ and $B$; and is the frequency. The loss function $L_{CL}$ that is caused by correntropy is described as follows:

$$L_{CL}(\hat{b}, b) = 1 - K_\beta\left(b - \hat{b}\right) \tag{8}$$

Here, the Gaussian kernel is used to determine the correlation. It can be rewritten as

$$L_{CL}(\hat{b}, b) = 1 - G_\beta\left(b - \hat{b}\right) \tag{9}$$

$$L_{CL}(\hat{b}, b) = exp\left(-\frac{\left(b - \hat{b}\right)^2}{\beta^2}\right) \tag{10}$$

Here, we propose a mixed loss function, FC-loss, defined as the product of focal loss and CL-loss.

$$L_{FCL}(\hat{b}, b) = \begin{cases} L_F(\hat{b}, b), & 0 < t \leq m \\ L_{CL}(\hat{b}, b), & 0 < t < M \end{cases} \tag{11}$$

where $t$ is the current time and $n$ is a fixed minimum requirement. After $n$ epochs of training with focal loss, we switch to CL-loss to finish off. The total number of epochs is denoted here by $M$. The CL-loss has numerous benefits, such as improved generalization and noise resilience. This is why we pretrain the model with focus loss across several iterations. To compensate for the class-weighting disparity, the focus loss has been implemented.

---

**Algorithm 1. The Algorithms for Glaucoma detection using the proposed model**

**Input:**

Training and Testing Dataset $D_i$, Learning rate $\propto$, Epochs $M$, Batch size $B$

**Output:**

Classification Glaucoma, Normal

1. **Preprocess** the images
2. **Divide** the dataset into training, validation, and testing sets
3. **Initialize** the VNet model to extract the feature
4. **for** each epoch 1 to M

    a. **Concatenate** the features from VNet and the transfer learning model

    b. Pre-trained layers' weights frozen before further training of the new fully connected layer

    c. Fine tune the model

    d. Train the model on the training with optimizer SGD/Adam and a suitable loss function FCL

$$L_{FCL}(\hat{b}, b) = \begin{cases} L_F(\hat{b}, b), & 0 < t \leq m \\ L_{CL}(\hat{b}, b), & 0 < t < M \end{cases}$$

6. **Evaluate** the trained model on the validation set to tune any hyperparameters
7. **Evaluate** the model on the testing set to report the final recital of the model.
8. **Analyze** the confusion matrix

## Dataset

The ACRIMA dataset, created by the Spanish Ministry of Economy and Competitiveness, focuses on glaucoma image categorization using Topcon TRC retinal camera images. These high-quality images, classified by specialists, form the basis for distinguishing between normal and glaucomatous samples. Similarly, the ORIGA dataset (*Orlando et al., 2020*; *Lin et al., 2020*), part of the SiMES project, features 650 manually segmented images from a study of 3,280 Malay individuals, annotated with CDR and glaucoma/health labels (*Diaz-Pinto et al., 2019*; *Zhang et al., 2010*). Additionally, our study analyzed the REFUGE challenge, involving 1,200 images, to concentrate on glaucoma classification and optic disc/cup segmentation. we resorted to the same pre-processing and data augmentation methods we employed before. In summary, a semantic segmentation network was used to first extract an area positioned on the optic nerve head from each raw fundus picture. After the optic nerve head was removed, each picture was automatically cropped to a square $224\times224$ pixels in size to be used as input in the DL model. Data augmentation procedures were used to improve the quantity and kind of variance in the training set before the DL model was trained.

## RESULTS

To assess our model, we established a robust experimental setup, utilizing an Intel E5-2620 CPU for its multiple cores and threads, essential for efficient data preprocessing and model training. Our experiments were further accelerated by an NVIDIA Tesla M2090 GPU, chosen for its parallel processing capabilities that are ideal for deep learning tasks.

---

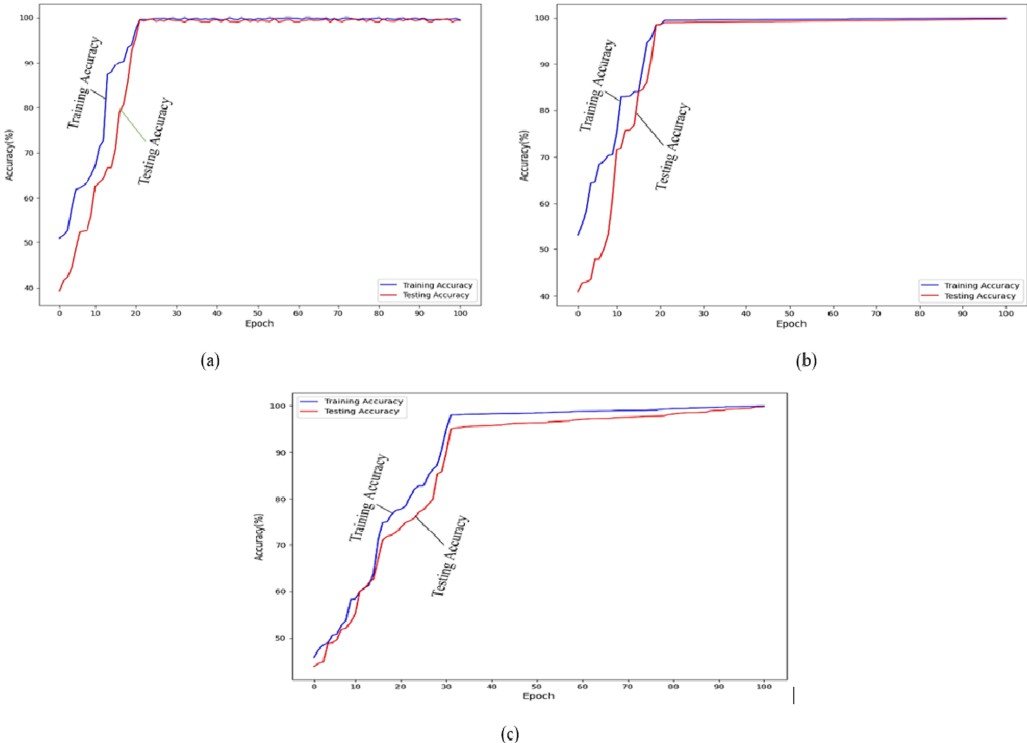

**Figure 1** **Prediction accuracy of the training process for our multi-task method.** (A) The accuracy of REFUGE; (B) the accuracy of the ACRIMA dataset; (C) the accuracy of the ORIGA dataset.

Python 3, known for its extensive machine learning libraries and user-friendliness, was our programming language of choice. We employed TensorFlow for its flexibility and efficiency in neural network training, coupled with the Keras library as a high-level API on TensorFlow, simplifying neural network construction and training. This setup enabled focused efforts on model architecture and hyper parameter optimization. Our technique's effectiveness was evaluated using metrics like accuracy, sensitivity, specificity, precision, F1-score, and area under the curve (AUC) on retinal images. Sensitivity and specificity measure the correct identification of glaucomatous and normal images, respectively. Accuracy reflects the overall correct identification rate, while the F1-score balances precision and recall. The ROC curve, depicted by the true positive and true negative rates, helps visualize performance, with the AUC indicating the model's predictive capacity.

The $F_{Pt}$ and $F_{Nt}$ both represent the normal picture that was misidentified as a glaucomatous one, and the glaucomatous image that was misidentified as a normal one. Figure 1 displays the dataset accuracy of the suggested model.

## RESULT AND DISCUSSION

Figure 1 displays the dataset accuracy of the suggested model.

The detailed loss of our proposed models on the dataset are shown Fig. 2. Our suggested deep learning architecture makes advantage of dropout in the two fully-connected layers.

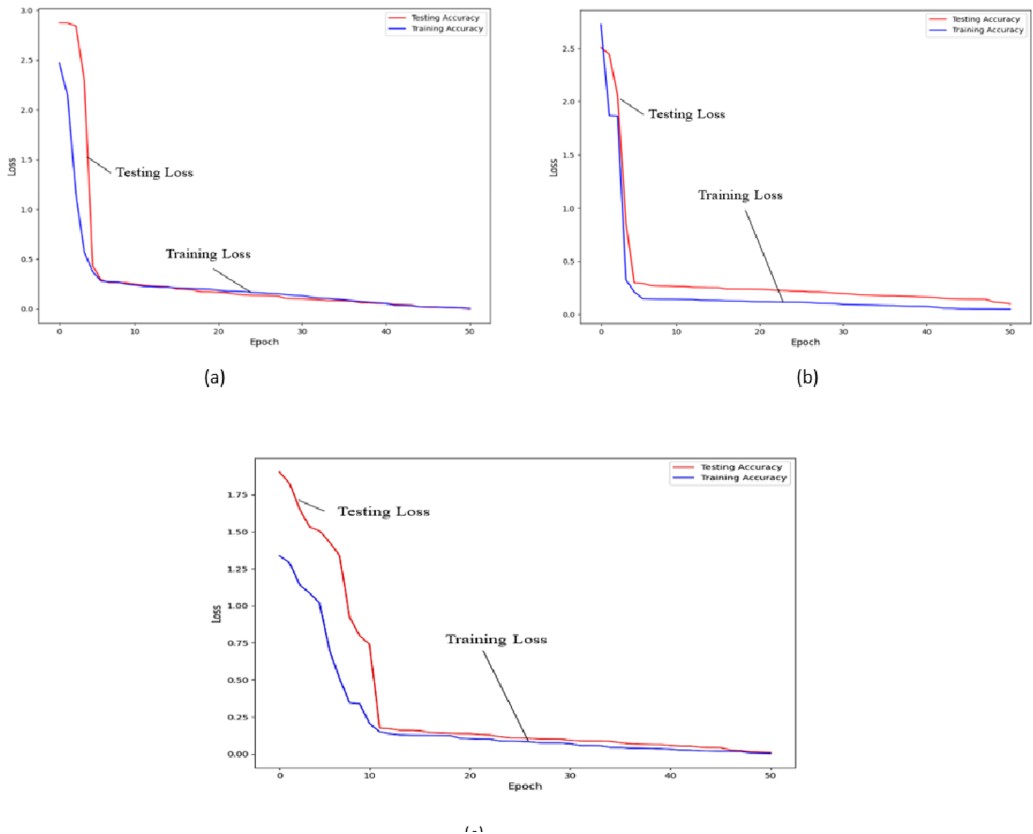

**Figure 2** **Prediction loss of the training process for our multi-task method.** (A) The loss of REFUGE; (B) the loss of the ACRIMA dataset; (C) the loss of the ORIGA dataset.

Every hidden neuron has a 50% chance of having its output reduced to zero. If neurons are removed, they cannot take part in the forward pass and will have no effect on the back propagation. We employ all the neurons and scale their outputs by a factor of 0.5 during testing.

In addition, a reliable glaucoma detection and classification framework will be able to distinguish between glaucomatous and healthy pictures. As a result, the true positive rate (TPR) is plotted in the confusion matrix to better illustrate the classification results. The acquired findings are displayed in Fig. 3, where it can be seen that the suggested technique displays a TPR of 0.99 for glaucoma-affected photos, demonstrating the efficacy of our approach. On top of that, our method achieves a mean glaucoma classification accuracy of 99.8%, 99.6% and 100% on the REFUGE, ACRIMA and ORIGA dataset. Because the ResNet V2 as the base network can compute a more accurate collection of image characteristics that better aid in identifying the unhealthy picture regions, our technique has a high degree of classification accuracy.

Taking into account all probable abnormalities in both the front and back parts of the eye, the suggested system sought to collect organized diagnostic information for forecasting of eye illnesses. Four deep learning algorithms' output is displayed in Fig. 3.

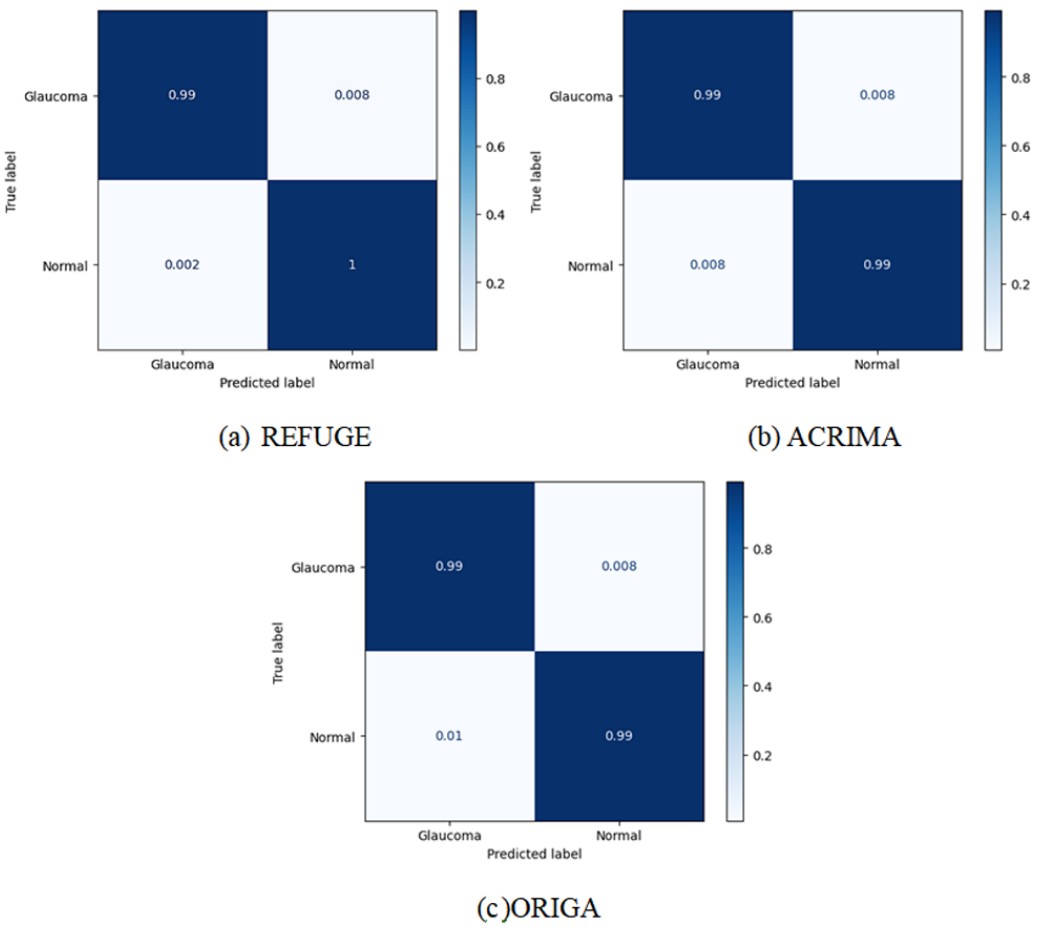

**Figure 3** **Confusion matrix of the training process for our multi-task method.** (A) The classification of REFUGE; (B) the classification of the ACRIMA dataset; (C) the classification of the ORIGA dataset.

Accuracy between 98.64% and 100% was attained when combining MTL with ALexNet, VGGNet, Xception, and InceptionResNetv2 models, with an average processing time of 3.7 s per picture. The MTL+InceptionResNetv2 combo model has the highest performance and accuracy. Consequently, we use the most effective pair as our glaucoma categorization criteria. Table 2 illustrates the performance evaluation of multiple deep learning models, including ALexNet, VGGNet, Xception, and InceptionResNetv2, on three distinct datasets: REFUGE, ACRIMA, and ORIGA. The evaluation metrics include accuracy, recall, area under the ROC curve (AUC), and F1-score, which collectively offer a comprehensive view of each model's diagnostic capabilities.

For the REFUGE dataset, ALexNet demonstrates strong performance with an accuracy of 98.9%, high recall of 0.99, impressive AUC of 0.98, and a robust F1-score of 0.98. Similarly, VGGNet achieves remarkable results with an accuracy of 99.0%, a recall of 0.98, a high AUC of 0.99, and a competitive F1-score of 0.98. Xception, while slightly lower in accuracy at 97.5%, maintains a recall of 0.97, an AUC of 0.97, and a commendable F1-score of 0.98. Notably, InceptionResNetv2 achieves flawless performance on the REFUGE dataset,

Peer J Computer Science

D et al. (2024), *PeerJ Comput. Sci.*, DOI 10.7717/peerj-cs.2186

**Table 2  Analyzing the efficiency of multiple learning methods.**

| Model/ Dataset | REFUGE | | | | ACRIMA | | | | ORIGA | | | |
|---|---|---|---|---|---|---|---|---|---|---|---|---|
| | Accuracy | Recall | AUC | F1-score | Accuracy | Recall | AUC | F1-score | Accuracy | Recall | AUC | F1-score |
| ALexNet | 98.9% | 0.99 | 0.98 | 0.98 | 99.5% | 0.99 | 0.99 | 0.99 | 99.2% | 0.99 | 0.99 | 0.99 |
| VGGNet | 99.0% | 0.98 | 0.99 | 0.98 | 99.2% | 0.99 | 0.99 | 0.98 | 99.1% | 0.98 | 0.98 | 0.99 |
| Xception | 97.5% | 0.97 | 0.97 | 0.98 | 97.2% | 0.97 | 0.97 | 0.98 | 98.0% | 0.97 | 0.98 | 0.97 |
| InceptionResNetv2 | 100% | 1.00 | 1.00 | 1.00 | 100% | 0.99 | 0.99 | 0.99 | 100% | 0.99 | 0.99 | 1.00 |

securing a perfect accuracy of 100%, a recall of 1.00, an AUC of 1.00, and an exceptional F1-score of 1.00. These results showcase the potential of deep learning models in accurately diagnosing glaucoma in the context of the REFUGE dataset.

Moving on to the ACRIMA dataset, ALexNet continues to excel with an accuracy of 99.5%, a recall of 0.99, a high AUC of 0.99, and an outstanding F1-score of 0.99. VGGNet also maintains strong performance with an accuracy of 99.2%, a recall of 0.99, an AUC of 0.99, and an impressive F1-score of 0.99. Xception achieves an accuracy of 97.2%, a recall of 0.97, an AUC of 0.97, and a competitive F1-score of 0.98. InceptionResNetv2 demonstrates exceptional performance with a perfect accuracy of 100%, a recall of 0.99, an AUC of 0.99, and a commendable F1-score of 0.99. These results underline the robustness of deep learning models in effectively diagnosing glaucoma within the ACRIMA dataset.

Finally, in the case of the ORIGA dataset, ALexNet maintains its strong performance, achieving an accuracy of 99.2%, a recall of 0.99, a high AUC of 0.99, and an impressive F1-score of 0.99. VGGNet also exhibits remarkable results with an accuracy of 99.1%, a recall of 0.98, an AUC of 0.98, and an outstanding F1-score of 0.99. Xception attains an accuracy of 98.0%, a recall of 0.97, a competitive AUC of 0.98, and a commendable F1-score of 0.97. InceptionResNetv2 delivers exemplary performance with a perfect accuracy of 100%, a recall of 1.00, an AUC of 0.99, and a remarkable F1-score of 1.00. These findings underscore the robustness of deep learning models in accurately diagnosing glaucoma across various datasets.

The VNet model's initial benefit over earlier segmentation methods is that it makes use of information about both global location and context simultaneously. There are several more benefits to using the VNet method. When compared to alternative methods for segmentation problems, the VNet model stands out for a number of reasons. Additionally, it enhances performance on tasks linked to segmentation using a minimal amount of training data. Possessing this is advantageous. To enhance representation learning in subsequent convolutions, in order to improve the system's accuracy, we up-sample its features and fuse them with more detailed feature maps generated by an encoder system.

Our study's promising results highlight the need to address key challenges for broader applicability. One major limitation is dataset variability, including differences in ethnicity representation, data collection methods, labeling standards, and image quality. The models, trained on specific datasets, showed varied effectiveness across different datasets, as retinal appearance can differ by ethnicity. Additionally, our datasets only included high-quality images, unlike real-world scenarios where low-quality and artifact-laden images are common. Therefore, creating a comprehensive screening dataset encompassing diverse ethnicities, genders, image qualities, and comorbidities is crucial for enhancing the model's practical effectiveness.

In our study, the AlexNet model demonstrated inference times for a single image ranging from a few milliseconds to about 20 ms, while VGGNet required 20 to 50 ms. The inference time for a single image using the Inception model ranged from 30 to 100 ms, and the ResNetv2 model took about 40 to 120 ms. Our deep learning-based strategy significantly improves the accuracy and reliability of glaucoma detection, with InceptionResNetv2 showing high performance metrics across multiple datasets. This

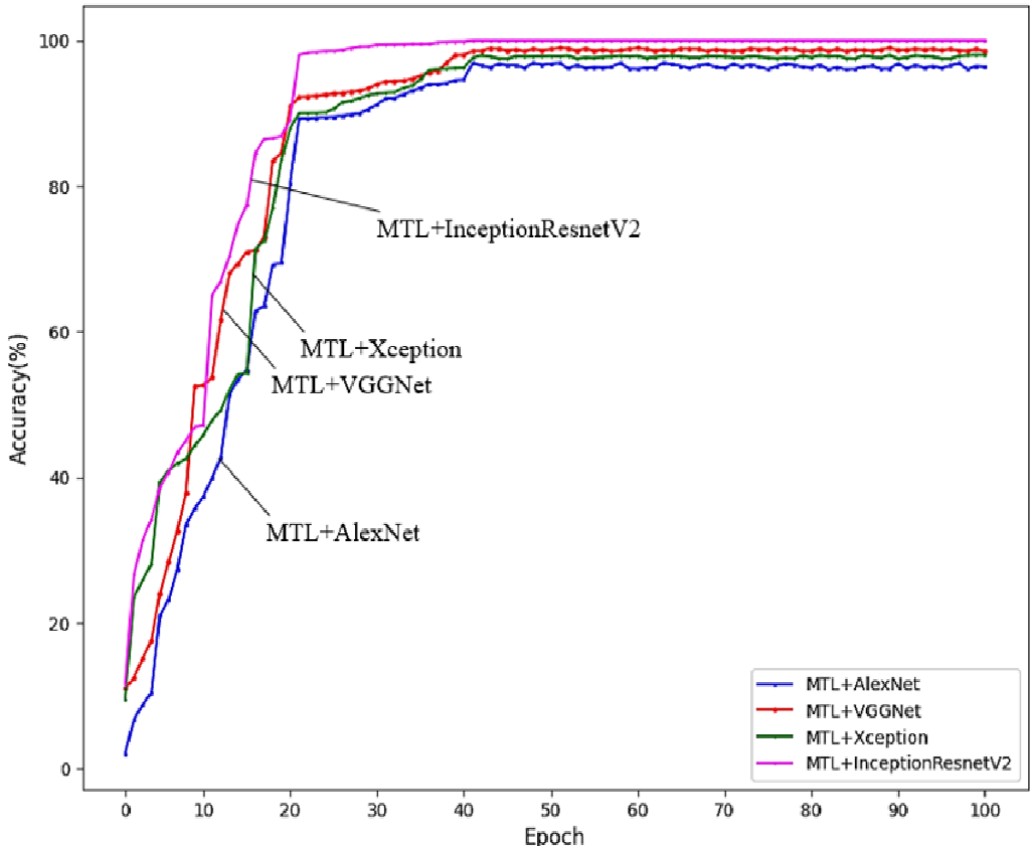

**Figure 4** Performance of various multi task learning algorithms.

automation in diagnosis, eliminating manual feature engineering, enhances efficiency and aids healthcare professionals in timely and accurate glaucoma identification.

## Challenges and limitation

Our research into diabetic eye disease detection using deep learning faced several complex challenges. Data preprocessing was intricate, especially with retinal image datasets, necessitating data augmentation, image normalization, and handling of imperfect data. Hyperparameter optimization also proved difficult, with choices in learning rates, batch sizes, and optimizers impacting model performance, requiring extensive experimentation for fine-tuning. The high computational demands of using advanced architectures like VGGNet and InceptionResNetv2 posed resource challenges, emphasizing the need for efficient GPU memory management and hardware utilization as in Fig. 4. Additionally, the risk of overfitting necessitated strategies like dropout and regularization to ensure our models' robustness in real-world applications. In our medical application study, interpretability was crucial. We tackled the challenge of creating transparent, interpretable model outputs to clarify decision-making processes, essential for clinical trust and adoption. Developing visualization and explanation methods was key to this transparency. Transitioning from development to real-world deployment also presented challenges,

including managing model size, ensuring real-time inference, and integrating with healthcare systems.

Our system, while impactful in diabetic eye disease detection, has limitations. Performance is tied to training data availability and diversity, highlighting the need for more extensive and varied datasets. Achieving fully interpretable outputs for informed clinical decisions remains challenging, requiring further research into advanced interpretability techniques. Additionally, system accessibility is limited by specific hardware requirements and substantial computational resources. Future work should explore model compression and optimization for wider accessibility. Finally, to ensure robustness and reliability, testing on diverse patient populations is necessary. Addressing these aspects is crucial for advancing the system's effectiveness, interpretability, and generalizability in clinical practice.

## CONCLUSIONS

The proposed study presents an advanced deep learning (DL) framework for multitask learning in glaucoma detection. This framework effectively discerns key features for accurate glaucoma diagnosis, demonstrating deep learning's prowess in analyzing retinal fundus images. A novel optic nerve head feature and a mixed loss function address class imbalance and outliers in complex datasets. The InceptionResNetv2 model shows exceptional accuracy (100%) on REFUGE, ACRIMA, and ORIGA datasets. Other architectures like AlexNet, VGGNet, and Xception also perform well, with high accuracy rates. Future research should focus on model generalization across diverse data and populations, enhancing resilience and accuracy through advanced image processing techniques. Investigating advanced data augmentation and transfer learning methods can overcome data size and diversity constraints. Improving interpretability of AI diagnostics is crucial for healthcare adoption. Additionally, exploring edge computing and optimization for real-world deployment is essential for broader accessibility and efficient operation in clinical settings.

### Funding

This work was supported by the Deanship of Research and Graduate Studies at King Khalid University through small group research under grant number RGP1/261/45. The funders had no role in study design, data collection and analysis, decision to publish, or preparation of the manuscript.

### Grant Disclosures

The following grant information was disclosed by the authors:
King Khalid University through small group research: RGP1/261/45.

### Competing Interests

The authors declare there are no competing interests.

## Author Contributions

- Banumathy D conceived and designed the experiments, prepared figures and/or tables, and approved the final draft.
- Swathi Angamuthu analyzed the data, authored or reviewed drafts of the article, and approved the final draft.
- Prasanalakshmi Balaji performed the experiments, prepared figures and/or tables, and approved the final draft.
- Mousmi Ajay Chaurasia performed the computation work, authored or reviewed drafts of the article, and approved the final draft.

## Data Availability

The datasets are available at Zenodo:

- Zhuo, Z., Shou Yin, F., Liu, J., Kee Wong, W., Meng Tan, N., Hai Lee, B., Cheng, J., Yin Wong, T., Fu, H., Li, F., IgnacioOrlando, J., & Bogunović, H. (2024). Diabetic Glaucoma(ORIGA,REFUGE,ACRIMA) [Data set]. Zenodo. https://doi.org/10.5281/zenodo.10674885.

- Revolutionizing Diabetic Eye Disease Detection. (2023). Revolutionizing Diabetic Eye Disease Detection. Zenodo. https://doi.org/10.5281/zenodo.10115028.

## Supplemental Information

Supplemental information for this article can be found online at http://dx.doi.org/10.7717/peerj-cs.2186#supplemental-information.

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
