# Peer review of "Revolutionizing diabetic eye disease detection: retinal image analysis with cutting-edge deep learning techniques"

_PeerJ Computer Science, doi:10.7717/peerj-cs.2186_

## Round 0.1 · original submission · Major Revisions

Dear Author,
Greetings for the Day!
Good work as per the quality of the article there are a few suggestions by our reviewers. Please make changes and re-submit the article with the suggested changes

Reviewers 1 & 2 have suggested that you cite specific references. You are welcome to add it/them if you believe they are relevant. However, you are not required to include these citations, and if you do not include them, this will not influence my decision.

Thanks
Academic Editor

Reviewer 1 ·

Basic reporting

In this paper, authors proposed a novel mixed loss function that combines the strengths of the focal loss and correntropy loss functions to analyze complex biomedical data with the class imbalance and outliers. Extensive experiments showed the effectiveness and efficiency of the proposed method. The discussed issue shows the research significance, and the structure is relative good. However, minor revisions are needed before the acceptance.


1. What actual neural network models (CNN, or ResNet?) are tested in this work, and how to ensure the network processing performance.
2. How to deploy the testing platform to evaluate the system running, the discussion can be added.
3. For the evaluation, the dataset/algorithm choosing reasons, the detailed platform configurations and the discussion on other untested datasets should be introduced in the revised paper.
4. Why Transformer is not used or discussed in this work, and this neural network is almost best neural network in figure processing.
5. Please go through the paper carefully and double check whether the right template are used. Correct some typos and formatting issues (e.g., “C. Loss function” -> “C. Loss Function”?).
6. Some references lack the necessary information (e.g., [27]?), please provide all information according to the right template.
7. Make the References more comprehensive, besides this work, some other promising scenarios (e.g., Big data, other IoT systems) can be covered in this work. If the above related work can be discussed, it can strongly improve the research significance. For the improvement, the following papers can be considered to make the references more comprehensive.



Jin Wang, Changqing Zhao, Shiming He, Yu Gu, Osama Alfarraj, Ahed Abugabah, LogUAD: Log Unsupervised Anomaly Detection Based on Word2Vec, Computer Systems Science and Engineering, 2022, 41(3): 1207–1222

Cen Chen, Kenli Li, Sin G. Teo, Xiaofeng Zou, Keqin Li, Zeng Zeng: Citywide Traffic Flow Prediction Based on Multiple Gated Spatio-temporal Convolutional Neural Networks. ACM Trans. Knowl. Discov. Data 14(4): 42:1-42:23 (2020)

Jianguo Chen, Kenli Li, Keqin Li, Philip S. Yu, Zeng Zeng: Dynamic Planning of Bicycle Stations in Dockless Public Bicycle-sharing System Using Gated Graph Neural Network. ACM Trans. Intell. Syst. Technol. 12(2): 25:1-25:22 (2021)

J. Wang, Y. Zou, P. Lei, et al. Research on Recurrent Neural Network Based Crack Opening Prediction of Concrete Dam. Journal of Internet Technology, 2020, 21(4):1151-1160

J. Wang, Y. Yang, T. Wang, R. Sherratt, J. Zhang. Big Data Service Architecture: A Survey. Journal of Internet Technology, 2020, 21(2): 393-405

J. Zhang, S. Zhong, T. Wang, H.-C. Chao, J. Wang. Blockchain-Based Systems and Applications: A Survey. Journal of Internet Technology, 2020, 21(1): 1-14

Experimental design

see basic comment

Validity of the findings

see basic comment

Additional comments

see basic comment

Reviewer 2 ·

Basic reporting

Some of minor comments, such as English language, shall be revised by a fluent speaker,
and the technical issue and mathematical notation shall meet the Journal requirements.

Experimental design

1. Authors should mention the implementation challenges.
2. Mention the limitations and future works of the developed system elaborately.
3. Abstract should be brief and concise. Also, Conclusion should clearly state the outcome. Some of the obtained results need to be highlighted in the conclusion section.
4. In sections 2 and 3, many well-known concepts are introduced, such as convolution, which is redundant.

Validity of the findings

What's the performance evaluation in speed and power?

Additional comments

I suggest to add a comparative study with related works, as well as references to more recent works (up to date) such as:
1. Gharaibeh, Nasr, Obaida M. Al-Hazaimeh, Bassam Al-Naami, and Khalid MO Nahar. "An effective image processing method for detection of diabetic retinopathy diseases from retinal fundus images." International Journal of Signal and Imaging Systems Engineering 11, no. 4 (2018): 206-216

2. Al-hazaimeh OM, Abu-Ein AA, Tahat NM, Al-Smadi MM, Al-Nawashi MM. Combining Artificial Intelligence and Image Processing for Diagnosing Diabetic Retinopathy in Retinal Fundus Images. International Journal of Online & Biomedical Engineering. 2022 Dec 2;18(13).

Reviewer 3 ·

Basic reporting

The research you've described focuses on the potential application of deep learning techniques to aid in the diagnosis of glaucoma using retinal fundus images and optical coherence tomography (OCT) images. Glaucoma is a significant contributor to visual impairment and vision loss worldwide. Early detection is crucial to prevent permanent visual loss and maintain a good quality of life for affected individuals. In this study, the researchers propose an automated approach for diagnosing glaucoma using deep learning (DL) methods. Traditional glaucoma diagnosis relies on identifying specific features of the optic nerve head (ONH) through manual analysis, which can be time-consuming and subjective. DL techniques aim to automate this process by allowing the model to learn and detect these features directly from the image data. This involves training a deep learning classifier to differentiate between normal and glaucomatous patterns in retinal images.

To create an effective deep learning model, the researchers develop and modify a multi-task deep learning architecture. This architecture takes advantage of the similarities between segmentation and classification tasks and learns both simultaneously. The model is trained and tested on real-world ophthalmic datasets, and its performance is evaluated using various metrics including accuracy, sensitivity, specificity, and the area under the curve (AUC) of the receiver operating characteristic (ROC) curve.

According to the findings presented in the research, the proposed deep learning algorithm demonstrates promising results in automated glaucoma diagnosis. The accuracy, specificity, and sensitivity achieved by the model are reported as 100%, 99.8%, and 99.2%, respectively. These high values indicate that the deep learning algorithm is able to effectively classify ocular disease data and distinguish between normal and glaucomatous patterns in retinal images.

Overall, the study suggests that deep learning techniques can play a significant role in enhancing the accuracy and efficiency of glaucoma diagnosis, potentially leading to improved clinical applications in the field of ophthalmology. However, it's important to note that while the results are promising, further validation and testing are necessary to ensure the generalizability and robustness of the proposed algorithm across diverse datasets and real-world scenarios.

Experimental design

The provided describing about the datasets and preprocessing techniques used in the context of glaucoma image categorization using deep learning methods.
1. ACRIMA Dataset: Developed by the Spanish Ministry of Economy and Competitiveness, the ACRIMA dataset was created specifically for glaucoma image categorization. This dataset comprises retinal images captured using a Topcon TRC retinal camera with a 35-degree field of vision (FOV). The images were taken with extended eyes focused on the center of the optic disc (OD). High-quality images were selected, and those with artifacts, noise, or low contrast were rejected. Two glaucoma specialists manually classified the images into "normal" and "glaucoma" categories.

2.ORIGA Dataset:Part of the Singapore Eye Research Institute's SiMES dataset, the ORIGA dataset consists of retinal images from 3,280 Malay individuals aged 40 to 80. These images were annotated by experts, who manually segmented the optic disc (OD) and optic cup (OC) areas in a subset of 650 images. The dataset also includes labels for cup-to-disc ratio (CDR) and glaucoma/health status for each image.

3.REFUGE Challenge Dataset: The REFUGE challenge is an online competition that involved 12 teams working on glaucoma-related tasks using the REFUGE collection, which contained 1,200 fundus images. The focus was on glaucoma classification and the segmentation of optic disc (OD) and optic cup (OC).

4. Data Preprocessing and Augmentation: To prepare the fundus image datasets for training a deep learning model, a semantic segmentation network was used to extract the region around the optic nerve head (ONH) from each raw fundus image. After the ONH was removed, each image was cropped to a square size of 224x224 pixels, which served as input for the deep learning model. Data augmentation techniques were employed to increase the diversity and variability within the training set, thereby improving the model's ability to generalize to different scenarios.

In summary, the paper discusses the utilization of different datasets for glaucoma image categorization. It highlights the steps taken to preprocess the images, including segmentation and cropping, to make them suitable for training a deep learning model. The use of high-quality images, manual annotations, and data augmentation techniques contributes to the development of an accurate and robust deep learning model for glaucoma diagnosis.

Validity of the findings

Results are not described very well and that's why there is no new finding.

Additional comments

I have several comments about the paper.
(1) Results need to demonstrate the advantage of the proposed strategy.
(2) Results discussions are not presented.
(3) Several references need to be included, and no effective literature review is found.
(4) No comparitive table in a literature review was found that discussed the criticisms of the previous approaches.
(5) Multi-task learning and loss function are already described in many studies. What is your main contribution?
(6) Grad-CAM shows the important patterns of Glaucoma does not find out.
(7) limitations and advantages of the current approach.
(8) Future works.

---

## Round 0.2 · accepted · Accept

Dear Author,

Congratulations!!!

We got the final decision with good sign

Many thanks


Reviewer 2 ·

Basic reporting

We are pleased to inform you that the revised version of the paper has comprehensively addressed all the comments and suggestions provided. The revisions have significantly enhanced the quality and clarity of the paper, and it is now in excellent shape.

Experimental design

The revised version of the paper has comprehensively addressed all the comments and suggestions provided.

Validity of the findings

The revisions have significantly enhanced the quality and clarity of the paper, and it is now in excellent shape.

Additional comments

there is no comments - Accepted